# Beyond Visual Interpretation: Quantitative Analysis and Artificial Intelligence in Interstitial Lung Disease Diagnosis “Expanding Horizons in Radiology”

**DOI:** 10.3390/diagnostics13142333

**Published:** 2023-07-10

**Authors:** Gaetano Rea, Nicola Sverzellati, Marialuisa Bocchino, Roberta Lieto, Gianluca Milanese, Michele D’Alto, Giorgio Bocchini, Mauro Maniscalco, Tullio Valente, Giacomo Sica

**Affiliations:** 1Department of Radiology, Monaldi Hospital, Azienda Ospedaliera dei Colli, 80131 Naples, Italy; roblieto@gmail.com (R.L.); giorgio.bocchini@gmail.com (G.B.); tullio.valente@gmail.com (T.V.); gsica@sirm.org (G.S.); 2Section of Radiology, Unit of Surgical Science, Department of Medicine and Surgery (DiMeC), University of Parma, 43121 Parma, Italy; nicola.sverzellati@unipr.it (N.S.); gianluca.milanese@unipr.it (G.M.); 3Department of Clinical Medicine and Surgery, Section of Respiratory Diseases, University Federico II, Monaldi Hospital, Azienda Ospedaliera dei Colli, 80131 Naples, Italy; marialuisa.bocchino@unina.it; 4Department of Cardiology, University “L. Vanvitelli”—Monaldi Hospital, 80131 Naples, Italy; mic.dalto@tin.it; 5Department of Pneumology Clinical and Scientific Institutes Maugeri IRCSS, 82037 Telese, Italy; mauromaniscalco@hotmail.com

**Keywords:** HRCT (high-resolution computed tomography), ILDs (interstitial lung diseases), AI (artificial intelligence)

## Abstract

Diffuse lung disorders (DLDs) and interstitial lung diseases (ILDs) are pathological conditions affecting the lung parenchyma and interstitial network. There are approximately 200 different entities within this category. Radiologists play an increasingly important role in diagnosing and monitoring ILDs, as they can provide non-invasive, rapid, and repeatable assessments using high-resolution computed tomography (HRCT). HRCT offers a detailed view of the lung parenchyma, resembling a low-magnification anatomical preparation from a histological perspective. The intrinsic contrast provided by air in HRCT enables the identification of even the subtlest morphological changes in the lung tissue. By interpreting the findings observed on HRCT, radiologists can make a differential diagnosis and provide a pattern diagnosis in collaboration with the clinical and functional data. The use of quantitative software and artificial intelligence (AI) further enhances the analysis of ILDs, providing an objective and comprehensive evaluation. The integration of “meta-data” such as demographics, laboratory, genomic, metabolomic, and proteomic data through AI could lead to a more comprehensive clinical and instrumental profiling beyond the human eye’s capabilities.

## 1. Introduction

Diffuse lung disorders (DLDs) and interstitial lung diseases (ILDs) represent a category of pathological conditions that manifest with widespread involvement of the lung parenchyma and interstitial network. From a purely classificatory point of view, they encompass a heterogeneous group of conditions that amount to approximately 200 distinct entities in the literature [1,2,3,4,5,6]. Radiology is increasingly integrated into the multidisciplinary diagnosis (MDD) and follow-up process of ILDs management thanks to high-resolution computed tomography (HRCT) of the chest, a rapid, repeatable, and essentially safe technique capable of providing highly accurate diagnostic information. It enables a refined detection of pulmonary abnormalities, allows for the evaluation of longitudinal changes during follow-up and resembles a low-magnification anatomical preparation from a histological point of view [7].

HRCT of the chest is a crucial tool for identifying distinctive patterns in DLDs and ILDs, playing a pivotal role in achieving an accurate diagnosis. Additionally, it provides valuable insights into alternative diagnostic possibilities and aids in characterizing mixed phenotypes, including the presence of small airway disease, comorbidities, and other factors contributing to progressive fibrotic patterns. The accurate interpretation of basic semiotic alterations observed during HRCT examination facilitates a comprehensive differential diagnosis in which the radiologist is called to express their judgment until a pattern diagnosis. This process requires high “skills” and, in agreement with the clinical and functional data, allows the multidisciplinary team (MDT) to often arrive at a confident diagnosis of pattern and finally of disease. In cases where the pattern-based diagnosis is not confident enough to reach a definitive diagnosis, such as an “indeterminate pattern” for usual interstitial pneumonia (UIP) according to idiopathic pulmonary fibrosis (IPF) guidelines or an “unclassifiable pattern” at the time of the initial diagnostic evaluation, the use of MDT is recommended. This is crucial for discussing atypical or extremely complex cases in order to achieve, at least in the early diagnostic phase, a “working diagnosis”, a procedure that, according to recent literature studies, can attain high levels of diagnostic confidence. MDT plays a pivotal role in the management of ILDs, serving as the gold standard for diagnosing various ILDs beyond IPF. This encompasses a broad spectrum of conditions, ranging from ILDs with autoimmune features (IPAFs: interstitial pneumonia autoimmune features) to fibrotic hypersensitivity pneumonia (f-HP) and non-specific interstitial pneumonia (NSIP). Nevertheless, the absence of definitive classification and standardized diagnostic criteria for certain entities poses a diagnostic challenge, particularly since a substantial number of inflammation-mediated ILD disorders may progress to fibrosis [5,8].

In the last decade, pulmonary fibrotic diseases have received a significant increase in attention, especially IPF, also due to the change in the diagnostic–therapeutic paradigm linked to the new guidelines for the management of fibrosing diseases, predominantly the progressive fibrotic phenotypes, and this focus has resulted in an ongoing search for a better profiling of fibrotic damage [9]. Currently, scientific societies are placing greater emphasis on understanding the evolutionary aspects of fibrotic diseases rather than solely focusing on strict diagnostic definitions. This shift is driven by the recognition that various forms of secondary ILDs, such as those associated with connective tissue diseases, exhibit progressive fibrotic damage that can significantly impact a patient’s prognosis, such as idiopathic forms. As a result, there is a pressing need for tools that can accurately and quantitatively assess changes in fibrotic damage, particularly in relation to therapeutic strategies and the use of antifibrotic drugs, which can effectively slow down functional decline. This approach holds significant relevance in clinical practice. The evolving dimensions of diagnostic evaluation align with the emerging ethical guidelines associated with personalized and precision medicine, which aim to provide tailored approaches for each patient. Within a comprehensive multidisciplinary framework, these refined aspects should encompass every clinical, laboratory, functional, and morphological facet. By integrating these elements, qualitative and quantitative insights can be obtained, enabling the adoption of targeted therapeutic strategies that optimize patient outcomes. The escalating utilization of quantitative software, including the integration of artificial intelligence (AI), presents a more objective and comprehensive approach to analyzing ILDs. This advanced analysis incorporates “meta-data” and holds the potential to extend beyond the limits of “human radiological vision”. The translational integration of multi-level data, encompassing demographics, laboratory findings, genomics, metabolomics, and proteomics, can further enrich the patient’s clinical and instrumental profiling [10,11,12,13,14,15,16]. The complexity of ILDs stems from the variable and occasionally unpredictable behavior of certain forms, particularly those characterized by a fibrosing pattern that may exhibit rapid disease progression. Therefore, the utilization of in vivo biomarkers, coupled with the increasing application of “liquid biopsy” techniques (such as blood and urine analysis), for genetic and molecular evaluations, represents an immensely intriguing field of study that holds significant potential for enhancing patient management [17,18,19]. In the realm of clinical and functional laboratory domains, morphological assessment through HRCT examination stands out for its practicality and reliability. However, limitations arise from inter-observer variability, diverse CT multidetector machines with varying protocols, and critical patient-related factors. These factors can significantly impact the accuracy of diagnostic judgments. The integration of objective analysis tools, such as quantitative CT (qCT) and AI, offers a unique opportunity to standardize the interpretation of anatomical data and mitigate inter-observer variability among human observers. These tools enable an objective analysis of lung abnormalities, including their extent, precise pattern identification, and evaluation of patients during follow-up. Ultimately, the goal is to obtain additional quantitative data that enhance the precision and objectivity of diagnostic assessments [20,21,22]. Quantitative imaging and AI are used to provide prognostic information for patients by comparing HRCTs and offering quantitative results on the progression of interstitial damage. These advanced diagnostic tools assist clinicians in assessing the potential response to treatment with anti-fibrotic drugs by correlating it with clinical and functional data [23,24]. Additionally, they empower radiologists to overcome the limitations of relying solely on subjective and visual evaluations. By utilizing quantitative analysis and AI algorithms, radiologists can extract precise and objective information from imaging studies, enabling them to identify subtle patterns, quantify disease progression, and assess treatment response accurately.

## 2. From the Realm of Semi-Quantitative Visual Analysis to the Realm of Pulmonary Quantification: A Groundbreaking Technological Metamorphosis

In previous studies, significant attention has been given to the analysis of pulmonary function in obstructive lung diseases, with a particular focus on quantifying damage from pulmonary emphysema and bronchial remodeling. However, the investigation of restrictive pathologies, such as fibrosing diseases, came later. Despite extensive research efforts to develop qualitative and semi-quantitative visual scoring systems to assess the extent of these diseases, the results have only partially met the initial expectations. Challenges such as inconsistent interpretations among different readers, both within and across individuals, as well as limited reproducibility, have impeded progress. Nonetheless, visual scores have remained a subject of evaluation alongside functional assessments, particularly in the study of primary or secondary forms of fibrotic conditions, capturing the scientific community’s attention for more than a decade since their introduction [25,26,27]. In order to overcome these limitations and improve objectivity, sensitivity, and reproducibility in the detection of ILDs, a range of computer-based quantification methods have been proposed. One approach to quantifying lung fibrosis involves measuring parameters such as mean lung attenuation (MLA), which represents the average attenuation value of lung parenchyma. Skewness, indicating the extent of histogram asymmetry, and kurtosis, reflecting the sharpness of the histogram peak, have also been utilized. The analysis of density histograms has shown promising results in studying patients with ILDs such as IPF and systemic sclerosis (SSc)-related ILD. These methods exhibit an enhanced sensitivity and reproducibility compared to visual evaluation, even when employing a low radiation dosage, thereby demonstrating an excellent performance [28,29,30,31,32].

The initial analyses primarily relied on two-dimensional evaluation methods conducted on a single slice or sequential CT imaging of the lung. The introduction of multidetector machines facilitated volumetric analysis. Extensive literature on these techniques has demonstrated a strong correlation, often in conjunction with the patient’s functional data. Consequently, the approach of “tissue density analysis” continues to hold significant value in both research and clinical settings, particularly for obstructive lung diseases such as emphysema and small airway evaluation. However, over time, its utilization has significantly expanded in the study of fibrotic diseases. This straightforward approach involves assessing the density of individual pixels or voxels in the CT scan (picture/volume elements) and generating a histogram that represents their distribution. Mathematical parameters derived from the density histogram, involving first-order statistical analysis, enable a quantitative characterization of pathological deviations from the “normal” lung parenchyma. These parameters capture variations in lung tissue density, whether in terms of a reduction or increase compared to the air component [33,34,35]. Undoubtedly, quantitative scoring required slightly more time compared to visual analysis. However, the software could be mastered in just 10–15 min for a complete evaluation of a patient, making it user-friendly. The key steps in the learning curve, in the past, involved uploading the CT scan and setting the region of interest (ROI). The ROI played a crucial role in recognizing the inherent densitometry of lung tissue, enabling the software algorithm to discern density variations for histogram calculations. Overall, this implied that effective software management is somewhat reliant on the operator’s expertise, but modern quantitative tools manage to segment the lungs almost perfectly without requiring additional help or correction from the operator; for example, two open-source tools such as image J (Image J, Java developed by the National Institutes of Health of the United States) and Slicer 3 D [36]. One notable strength of this approach is that lung density histograms can be easily performed using these open-source tools (Figure 1 and Figure 2).

Advanced lung densitometric analysis tools have undergone significant advancements, encompassing a wide range of additional functionalities that effectively address various requirements. These tools now incorporate features such as improved segmentation techniques, comprehensive lung analysis capabilities, a precise assessment of lung lesions, computer-aided detection (CAD) analysis, accurate airway examination, and volumetric 3D reconstruction for pre-surgical planning purposes. While densitometric analyses provide valuable assistance in objectively quantifying fibrosing lung damage in both primary and secondary forms of ILDs, their limitation lies in the simplification of the method, which may overlook subtle intrinsic density variations associated with subtle alterations. Instead, it relies on an overall assessment, resulting in a “cumulative densitometric vision” where the HRCT’s ability to provide precise and detailed anatomical evaluation is diminished. For instance, different ILDs with a distinct etiology, pathogenesis, and distribution of elementary lesions may exhibit similar density histogram values, and this resemblance is particularly evident in decreased kurtosis and increased asymmetry, which are associated with fibrosing damage. The conclusive result determines a significant limitation in the differential diagnosis of diseases with a similar fibrotic matrix but with different evolutionary behaviors.

## 3. Unlocking the Potential: Artificial Intelligence Revolutionizes Interstitial Lung Disease Diagnosis with Quantitative Imaging and Advanced Data Analysis

AI is a novel term used to describe computer systems able to solve specific tasks that commonly require human intelligence. AI is revolutionizing the field of ILD diagnosis through the integration of quantitative imaging and advanced data analysis techniques. By leveraging AI algorithms, researchers and clinicians can unlock the full potential of medical imaging data, enabling a more precise and accurate detection, classification, and prognosis of ILDs. This cutting-edge approach combines machine learning, deep learning, neural networks, and radiomics, empowering healthcare professionals with powerful tools to enhance diagnostic accuracy and optimize treatment strategies for ILD patients. Machine learning, an integral part of AI, revolves around the concept of computer systems adapting and learning from data samples to execute specific tasks. Unlike traditional programming methods with explicit rules and instructions, machine learning algorithms are designed to be trained or fitted using specific datasets. Among the AI techniques, supervised and unsupervised learning stand as powerful tools, each offering a unique perspective in unraveling the mysteries of ILDs. “Supervised Learning”: by utilizing labeled training data, this approach enables AI models to learn patterns and associations, ultimately aiding in disease classification and prediction. Through a process of meticulous training and validation, supervised learning algorithms acquire the ability to accurately identify specific ILD subtypes, such as IPF or HP, based on defined features and characteristics. This enables clinicians to make informed decisions regarding treatment strategies and prognostic evaluations, elevating patient care to unprecedented levels of precision. “Unsupervised Learning”: on the other hand, this approach serves as a beacon in unveiling the hidden patterns within ILDs. Without the need for predefined labels, unsupervised AI models excel at discovering intrinsic structures and relationships within complex ILD datasets. By applying advanced clustering and dimensionality reduction techniques, these models can unravel novel disease subtypes and identify intricate patterns that may elude human observation. Unsupervised learning empowers researchers to explore the vast landscape of ILDs, potentially uncovering new insights, biomarkers, and novel avenues for targeted therapies. While supervised and unsupervised learning differ in their methodologies, they are not mutually exclusive. In fact, their synergy holds the key to unlocking a deeper understanding of ILDs. By combining the strengths of both approaches, AI models can leverage the meticulous classification capabilities of supervised learning while simultaneously exploring the uncharted territories of unsupervised learning; this holistic approach not only enhances diagnostic accuracy but also opens doors to personalized treatment strategies, early detection, and improved patient outcomes. Achieving precise and clinically valuable algorithms in machine learning necessitates the utilization of suitable AI computational analysis and the incorporation of pertinent outcomes or ground truth. Powerful computing processors and machine learning methods were introduced by researchers, able to analyze volumetric data and to extract by CT scans image features and other informatics information on the densitometric variations on tiny pulmonary areas in order to evaluate diffuse lung disorders with the possibility of obtaining, also with colorimetric regional lung variations, a subtle difference between the HRCT areas (for example, normal lung, emphysema, GGO, consolidations, reticulations, honeycombing). These computational analyses, also called adaptative multiple-feature methods in a lung texture analysis, can provide “intelligent” maps of pulmonary morphological and densitometric variations, associated with an almost perfect computerized analysis of lung damage, to obtain distinct features for classifying different regional areas in a CT image. Machine learning models can also assist in ILD prognosis by analyzing a multitude of clinical and imaging variables to predict disease progression, survival outcomes, and treatment response. These models can integrate diverse datasets, including longitudinal imaging data, pulmonary function tests, genetic markers, and clinical features, to generate personalized prognostic assessments for ILD patients. Such prognostic tools can aid in treatment decision making and facilitate the development of tailored management plans. In the context of ILD management, machine learning algorithms can also contribute to the development of computer-aided systems for the automated detection and segmentation of ILD-related abnormalities on radiographic images. By automating the identification of specific lung patterns and lesions, these algorithms can improve efficiency, reduce inter-observer variability, and provide quantitative measurements of disease extent and progression. Therefore, the subsequent evolution of advanced pulmonary analysis techniques after “the lung tissue density analysis” has involved the introduction of “texture analysis”, which refers to a set of methods and algorithms for the extraction of information regarding the structural characteristics of an image and is also capable of extrapolating and evaluating different groups of radiomics parameters. This analysis can involve various approaches, including traditional feature engineering methods such as Gabor filters or texture co-occurrence matrices, as well as more advanced techniques such as machine learning or deep learning. One of the most well-known, effective, and widely used software applications for texture analysis is Computer-Aided Lung Informatics for Pathology Evaluation and Rating (CALIPER) [37]. This software, an example of machine learning trained by thoracic radiologists, utilizes a combination of volumetric histograms signature mapping features and automatically assigns each pixel to one of seven specific parenchymal patterns: normal, ground-glass opacity (GGO), reticular pattern, honeycombing, and low-density areas (mild, moderate, or severe); furthermore, it allows for the identification of additional and previously undetected “in vivo” biomarkers for improved and more effective patient phenotyping and profiling. In recent literature, an increasing body of evidence has demonstrated a robust correlation between CALIPER’s findings and functional tests, overall survival, and decline in lung function among patients diagnosed with IPF and other fibrotic ILDs, surpassing the performance of visual scoring methods. The application of textural analysis in ILD and other pulmonary conditions has shed light on the significance of quantifying vascular texture and perivascular abnormalities. The intricate network of pulmonary blood vessels undergoes dynamic changes during the progression of ILD. Tracking and segmenting lung vessels have proven to be relatively straightforward compared to other components such as airways and airspaces, particularly in healthy lungs. It is important to note that pulmonary diseases often involve abnormalities affecting multiple lung components, including airways, airspaces, interstitium, and vessels. Therefore, investigating the vascular component offers valuable insights into the impact of ILD on pulmonary structures, even during the early stages. Computer-based analysis of vascular structures has surpassed the limitations of visual interpretation, enabling the quantification of vascular complexity through a comprehensive assessment of anatomical representation. Furthermore, studying the intricate interaction between vessels and pulmonary tissue provides additional insights into disease behavior and treatment response. Consequently, the development of predictive models utilizing machine learning techniques for specific “textures” appears to be highly promising soon [38,39,40]. 

## 4. Unleashing the Potential of AI: Unraveling ILDs Mysteries through Deep Learning, CNN, Radiomics, and Lung Shrinkage

### 4.1. Deep Learning

Deep learning, a subfield of machine learning, has gained significant attention in recent years due to its remarkable ability to learn hierarchical representations from complex data. In the context of ILDs, deep learning techniques have shown great potential in several applications, revolutionizing the field of ILD research and management [41,42]. One of the primary applications of deep learning in ILDs is in the automated analysis and interpretation of medical images, particularly HRCT scans. Deep learning models, such as convolutional neural networks (CNNs), can be trained on large datasets of annotated HRCT images to automatically detect and classify various ILD patterns and abnormalities. Numerous studies have investigated the utilization of advanced imaging techniques and AI for the prediction and diagnosis of histopathologic conditions such as UIP. For example, a study introduced a CNN that utilized virtual wedges of the peripheral lung on HRCT to predict UIP [43]. CNN demonstrated moderate agreement with expert radiologists. In a more recent study, a DL model trained on a dataset of pathologically proven ILD was employed. The findings showed that the DL model outperformed visual CT analysis in predicting the histopathologic diagnosis of UIP and exhibited a higher reproducibility compared to expert radiologists. Specifically, when classifying cases as probable UIP based on a guideline, the DL model achieved a higher specificity compared to expert radiologists [43,44,45]. These models can learn to identify subtle radiological features indicative of specific ILD subtypes, including honeycombing, ground-glass opacities, reticulation, and traction bronchiectasis [12,13,46]. DL-based automated lung CT volumetry and fibrosis scoring have been shown to correlate with functional data and provide insights into the prognosis of IPF. DL algorithms have demonstrated a superior performance compared to thoracic radiologists in ILD classification and predicting survival outcomes. DL models have also outperformed experts in predicting histopathologic diagnoses and shown a better reproducibility. Additionally, the DL quantification of ILD patterns and extent has improved disease characterization and correlated well with functional data [12]. The deep texture analysis (DTA) provided by deep learning algorithms can aid radiologists in accurate and efficient ILD diagnosis and classification and in the prediction of disease progression and treatment response with HRCT; it is trained to distinguish fibrosis by utilizing image regions identified by radiologists as exhibiting normal lung parenchyma and typical patterns of fibrotic features. Representative regions labeled as reticulation, honeycombing, or traction bronchiectasis are employed to define the fibrosis category. In a sliding window manner, the algorithm classifies local regions within axial sections as either normal lung or fibrosis, which are identified through a separate segmentation process applied to the lung fields. The DTA fibrosis score is computed as the percentage of the total number of window regions classified as fibrosis (Figure 3A,B).

Deep learning models can integrate longitudinal imaging data, clinical variables, and other relevant biomarkers to generate predictive models; by capturing complex relationships and temporal dynamics within the data, these models can provide valuable prognostic information for ILD patients [14,47,48]. Additionally, deep learning models can help to identify patients who are likely to respond positively to specific treatments, facilitating personalized therapeutic strategies and being valuable for ILD risk stratification and early detection. By utilizing extensive datasets including health records, genetics, and environmental factors, deep learning models can identify individuals at a higher risk of ILD development. Early detection is crucial for timely intervention and improved outcomes. These models could also help in identifying high-risk individuals and facilitating targeted screening. However, challenges exist in applying deep learning to ILDs: large and diverse datasets are needed for training, which may be limited for rare or specific subtypes. It is important to note that AI in general and deep learning specifically in this context are designed to assist radiologists rather than replace them. The primary objective of the software is to streamline the interpretation process, alleviate the workload, and enhance the accuracy and consistency of ILD diagnoses. Radiologists, in collaboration with clinicians, can leverage the insights and recommendations provided by the AI tool to make well-informed clinical decisions. As with any AI tool, the performance of deep learning algorithms relies on the quality and diversity of the training data that they have been exposed to and, for these reasons, the continuous validation and refinement of these algorithms are critical to ensure their effectiveness and generalizability across different patient populations. In conclusion, AI software utilizing deep learning techniques serves as a valuable aid to radiologists in the study of ILDs. By functioning as a pattern classifier, it assists in the analysis and interpretation of HRCT scans, providing automated annotations, disease quantification, and diagnostic suggestions. However, it is essential to recognize that human expertise and judgment remain integral to the diagnosis and management of ILDs [49].

### 4.2. Convolutional Neural Network, Radiomics, and Lung Shrinkage

The introduction of CNNs has ushered in a new era of diagnostic precision in ILDs. Leveraging their ability to extract complex features from medical images, CNNs have redefined the landscape of ILD diagnostics. Concurrently, radiomics has empowered clinicians to delve deeper into the quantitative analysis of ILD radiographs, enabling a comprehensive characterization and classification of these complex diseases. Integrated algorithms incorporating clinical assessment, functional tests, and CT imaging, along with radiomics-based features, have shown promise in evaluating and predicting prognosis in patients with fibrotic ILDs. By extracting 26 radiomic features from routine chest CT scans, these algorithms provide valuable information for predicting progression-free survival in individuals with SSc-ILD. The integration of radiomics enhances prognostic evaluation and enables more informed treatment decisions for improved patient care [50]. 

CNNs in the field of thoracic imaging have proven to be a powerful tool for automated image analysis in ILDs. By leveraging their capacity to capture subtle patterns and textures within high-resolution radiographic data, CNNs surpass human visual perception, enabling a superior detection and classification of ILD subtypes [16,51]. From distinguishing IPF from other ILDs to predicting disease progression, CNNs offer a multifaceted approach that aids in both diagnosis and prognosis. Furthermore, the integration of transfer learning improves the CNN performance, underscoring their versatility in ILD research. Radiomics, an emerging field within medical imaging, complements CNNs by extracting an extensive array of quantitative imaging features from radiological images. These features encompass a wide range of morphological, textural, and statistical descriptors, providing a holistic representation of disease characteristics [52]. Leveraging advanced machine learning algorithms, radiomics models can stratify ILDs, differentiate between disease stages, and even predict treatment response [53]. By unraveling hidden imaging biomarkers, radiomics demonstrates its potential as a non-invasive and objective tool for ILD assessment. The integration of CNNs and radiomics represents a paradigm shift in the management of ILDs for both primary and secondary forms, such as connective tissue diseases [50,54]. Together, they offer a comprehensive and detailed understanding of ILDs, facilitating accurate diagnosis and personalized treatment plans. CNNs excel at extracting complex visual features from radiological or medical nuclear data, while radiomics enables a quantitative assessment of disease characteristics. The synergy between these two approaches empowers radiologists and clinicians to uncover previously unrecognized patterns and correlations, leading to an improved diagnostic accuracy and prognostic capabilities. As CNNs and radiomics continue to evolve, their impact on ILD diagnosis and management is expected to grow exponentially. The development of large-scale, curated datasets will further enhance the performance and generalizability of CNN models. Moreover, the integration of multi-modal imaging data, such as computed tomography (CT) and positron emission tomography (PET), holds great promise in unraveling the complexities of ILDs [55,56].

Lastly, new additional methods of lung evaluation using advanced AI techniques have emerged as additional tools for integrating clinical and pulmonary functional data. One of these methods is the assessment of the so-called “lung shrinkage”, a key component of worsening lung fibrosis in ILD, which could be effectively assessed using advanced imaging techniques such as CT. The regional distribution of lung shrinkage in ILD typically starts in the lower peripheral regions of the lungs, gradually ascending to the upper apical regions. This pattern may be attributed to mechanical stress on the alveolar epithelium and the fibroproliferative response. For this reason, the measurement of lung shrinkage using elastic registration and deep learning classifiers provides spatial information about the deformation process, enhancing our understanding of disease progression. It may also assist in the early detection and monitoring of ILD. However, it is important to consider lung shrinkage in conjunction with other markers, such as changes in lung function parameters such as forced vital capacity (FVC) and the diffusing capacity of carbon monoxide (DLCO), to obtain a comprehensive assessment of disease severity and treatment response. By combining these approaches, including advanced imaging techniques, quantitative analysis, and the evaluation of lung function, a more holistic understanding of lung shrinkage in ILD can be achieved, enabling an improved monitoring and management of this complex condition [15].

## 5. Conclusions

The application of AI techniques in the field of ILDs has revolutionized thoracic radiology, opening new possibilities. The transition from quantitative imaging to machine learning, deep learning, neural networks, radiomics, and lung shrinkage has introduced promising tools for enhancing the diagnosis, characterization, and prognosis of ILDs. Machine learning algorithms, with their ability to analyze extensive datasets and identify intricate patterns, have demonstrated potential in automating classification and prediction tasks for ILDs. Through training on annotated datasets, these algorithms can learn to recognize disease-specific features and assist in distinguishing between different ILD subtypes. Deep learning, a subset of machine learning, has emerged as a powerful technique in ILD analysis. CNNs, a type of deep learning architecture, have shown remarkable performance in automated image segmentation, allowing for the precise delineation of lung abnormalities and accurate quantification of disease burden. Furthermore, deep learning models can extract high-level features from medical images, capturing subtle nuances that may be imperceptible to the human eye. Radiomics, another exciting field, involves extracting numerous quantitative features from medical images, enabling the creation of comprehensive imaging biomarkers. By leveraging radiomics, researchers have identified imaging signatures associated with specific ILDs, leading to an improved disease characterization and potentially facilitating treatment selection and prognosis assessment. Lung shrinkage, a technique involving lung deflation to eliminate confounding factors such as blood vessels, has been combined with AI methods to enhance the accuracy of ILD analysis. By reducing anatomical distortions and improving the alignment of corresponding image features, lung shrinkage can enhance the performance of automated algorithms, resulting in more reliable and reproducible results. Despite the immense potential of these advanced techniques, it is crucial to acknowledge their limitations. Challenges such as dataset heterogeneity, a lack of standardization in imaging protocols and annotations, the interpretability of AI models, and the need for rigorous clinical validation must be addressed.

The current landscape of AI in ILDs primarily focuses on lung lesion detection systems, particularly in the context of the COVID-19 pandemic. However, the application of AI in ILDs remains relatively limited. Research efforts have predominantly concentrated on the pattern detection, quantification, diagnosis, and prognosis of ILDs using imaging techniques. Nonetheless, there are several challenges hindering the development of AI algorithms for ILDs. These challenges include the rarity of ILDs, the wide range of entities causing pulmonary involvement, the scarcity of structured data, data governance issues, limited access to clinical information, and a shortage of annotated datasets. Features such as FVC have been used for predictions, and prognosis inference remains challenging for both physicians and AI systems. Utilizing longitudinal data from HRCT scans and pulmonary function tests shows promise but requires further research. Although radiomic features have shown success in diagnostic tools and the prognosis of SSc-ILD, research in the prognosis of IPF using radiomics is limited. Efforts have been made to address these challenges by creating open-access databases to facilitate data access for researchers. However, the accessibility of the required data remains a potential hurdle for future research in AI-driven chest radiology [57]. Additionally, ethical considerations including patient privacy and data security, as well as the importance of close collaboration between radiologists, pulmonologists, and data scientists, must be emphasized to ensure the responsible and effective implementation of AI in routine clinical practice. In summary, the integration of AI techniques with radiological expertise holds significant potential in advancing ILD management. By harnessing the power of machine learning, deep learning, neural networks, radiomics, and lung shrinkage, we can deepen our understanding of ILDs, enable earlier and more accurate diagnoses, and ultimately improve patient outcomes. However, it is essential to carefully address the challenges and limitations associated with these techniques while maintaining a patient-centric and ethically responsible approach.

## Figures and Tables

**Figure 1 diagnostics-13-02333-f001:**
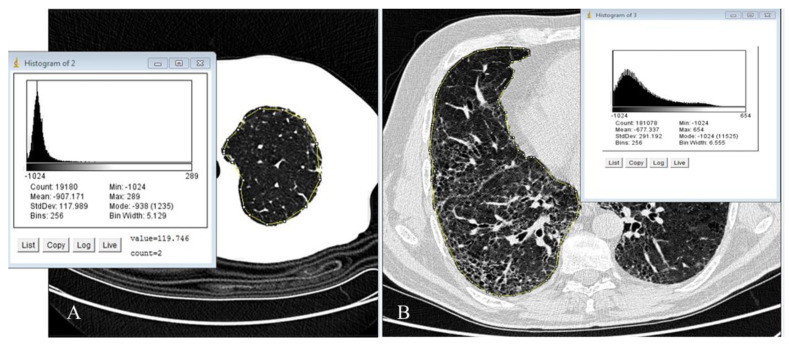
(**A**,**B**) (image J, Java open source): In (**A**), a control normal lung is depicted by a sharply shifted Gaussian curve with a narrow and tall peak. In (**B**), a representative slice of the right lung from a patient with advanced idiopathic pulmonary fibrosis, obtained through thin-section volumetric CT, is shown and illustrates the density histogram of the fibrotic lung, which exhibits a less pronounced peak and skewness compared to panel A. The segmented area of interest is highlighted in yellow. On the right side, the results of the digital processing analysis for this specific slice are presented. Once the sampling of the entire lung was completed, the software automatically generated averaged data from the analysis of all slices of both lungs.

**Figure 2 diagnostics-13-02333-f002:**
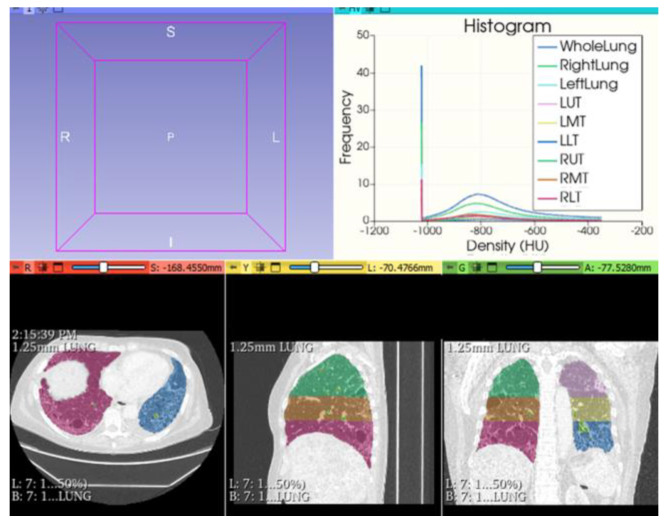
(Slicer3D): UIP/IPF: Color-coded map differentiated for lung areas. Histogram analysis provides an accurate description, both globally and regionally, of mean lung density (MLD), skewness, and kurtosis parameters, with increased potential to stratify fibrotic damage.

**Figure 3 diagnostics-13-02333-f003:**
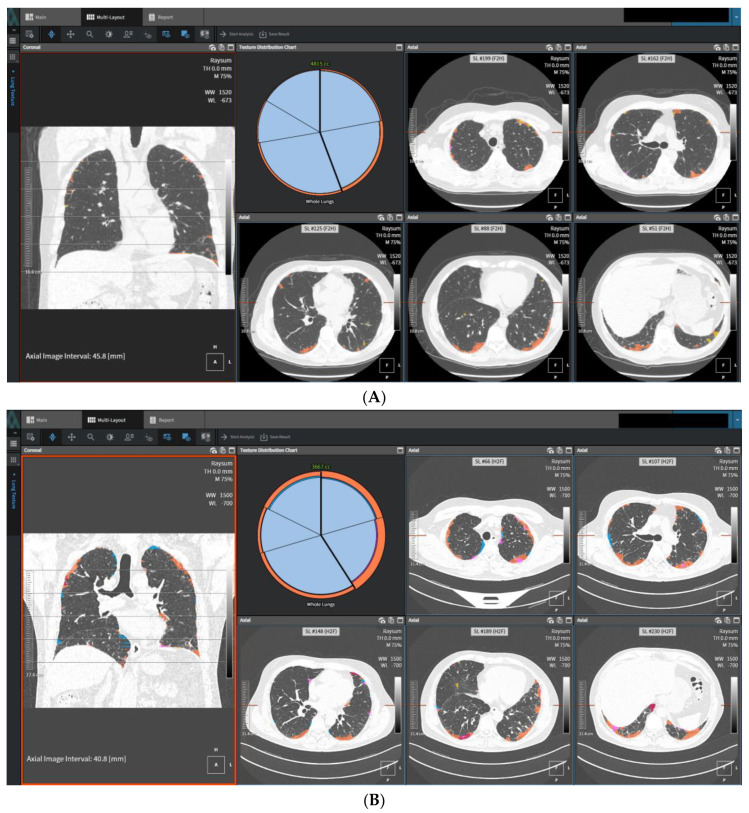
High-resolution computed tomography (HRCT) images and regions classified as fibrosis by data-driven texture analysis of a 53-year-old man former smoker: (**A**) the baseline of fibrosis score was 11.2%; baseline FVC % pred was 73%, *D*_LCO_ % pred was 69.0%. (**B**) HRCT images in the same subject at a nominal 78-week follow-up. Regions classified as fibrosis by DTA are shown in orange. The DTA fibrosis score increased by 12.6% percentage points at follow-up. FVC declined 10.5% (relative to baseline), *D*_LCO_ declined 13.0% (relative to baseline).

## Data Availability

Not applicable.

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
