# Peer review of "Beyond Visual Interpretation: Quantitative Analysis and Artificial Intelligence in Interstitial Lung Disease Diagnosis “Expanding Horizons in Radiology”"

_diagnostics, 2023, doi:10.3390/diagnostics13142333_

Round 1
Reviewer 1 Report
Nice and concise review of AI in interstitial lung diseases. However, the authors should discuss with examples how machine learning and deep learning have enabled differentiation and prognostication of ILDs. The introduction ends abruptly. Please review and revise as appropriate. Challenges in practical implementation of these newer techniques should be addressed in greater details. The authors can refer to more recent articles on the subject for a comprehensive review. For example- Recent Advancements in Computed Tomography Assessment of Fibrotic Interstitial Lung Diseases - PubMed (nih.gov) and Artificial Intelligence and Interstitial Lung Disease: Diagnosis and Prognosis - PubMed (nih.gov)
The authors should review the manuscript for syntax and grammatical errors. Some sentences are very long, such as in the Introduction section lines 45 to 48 and lines 66-70. These should be made shorter for easy reading. Overall, the quality of English language is acceptable and easy to understand.
Author Response
The "point by point" responses to the reviewer have been provided in the attached word document. Thank You.

Reviewer 2 Report
Re: Beyond visual interpretation: Quantitative Analysis and Artificial Intelligence in Interstitial Lung Disease Diagnosis “Expanding Horizons in Radiology”
This review article reports the summary of quantitative analysis and AI in ILD diagnosis. It should be noted that this review does not introduce specific quantitative analysis technics in detail, but rather describes the current trend of quantitative analysis in abstract terms. Reviewer considers this a clear review article to understand the trend of quantitative analysis.
Minor revision
・Please correct many mistakes in the manual when using abbreviations. For example, GGO (line237) from the first time is listed as an abbreviation, and GGO (line265) from the second time is listed with full spell. IPF is listed with full spell and as an abbreviation in two places. Some abbreviations are used for terms used only once in the text.
・MDT and MDD are used interchangeably and the meaning of the text is different in some parts. Please correct.
Author Response
The "point by point responses to the reviewer have been provided in the attached word document. Thank You.

Round 2
Reviewer 1 Report
1. Please remove the word team from line 57 as it is self-implied by MDT.
2. Please expand IPAD in line 62.
3. Please provide a citation for the study quoted in lines # 301-303.
Author Response
The requested considerations and corrections are attached in a word document for the reviewer's reference. Thank you
